# Structure-Based Virtual Screening towards the Discovery of Novel ULK1 Inhibitors with Anti-HCC Activities

**DOI:** 10.3390/molecules27092627

**Published:** 2022-04-19

**Authors:** Yang Gao, Ziying Zhou, Tingting Zhang, Situ Xue, Ke Li, Jiandong Jiang

**Affiliations:** Institute of Medicinal Biotechnology, Chinese Academy of Medical Sciences & Peking Union Medical College, Beijing 100050, China; yxgyang@126.com (Y.G.); zhouziying@imb.pumc.edu.cn (Z.Z.); zhangtingting@imb.pumc.edu.cn (T.Z.); xuesitu@imb.pumc.edu.cn (S.X.); like@pumc.edu.cn (K.L.)

**Keywords:** autophagy, hepatocellular carcinoma, ULK1, virtual screening

## Abstract

There is an urgent need to develop new effective therapies for HCC. Our previous study identified ULK1 as the potential target for HCC therapy and screened the compound XST-14 as a specific inhibitor of ULK1 to suppress HCC progression. However, the poor manufacturability of XST-14 impeded the process of its clinical translation. In this study, we first generated pharmacophore models of ULK1 based on the X-ray structure of UKL1 in complex with ligands. We then screened the Specs chemical library for potential UKL1 inhibitors. By molecular docking, we screened out the 19 compounds through structure-based virtual screening. Through CCK8 activity screening on HCC cells, we found that ZZY-19 displayed obvious cell killing effects on HCC cells. SPR assay indicated that ZZY-19 had a higher binding affinity for ULK1 than XST-14. Moreover, ZZY-19 induced the effects of anti-proliferation, anti-invasion and anti-migration in HCC cells. Mechanistically, ZZY-19 induces autophagy inhibition by reducing the expression of ULK1 on HCC cells. Especially, the combination of ZZY-19 with sorafenib synergistically suppresses the progression of HCC in vivo. Taken together, ZZY-19 was a potential candidate compound that targeted ULK1 and possessed promising anti-HCC activities by inhibiting autophagy.

## 1. Introduction

Primary liver cancer is a global health problem. Previous studies have shown that liver cancer remains the fifth most common malignancy in men and the eighth in women worldwide [1,2]. Hepatocellular carcinoma (HCC) is the most common type of primary liver cancer and is characterized by both phenotypic and molecular heterogeneity [3,4]. Although in the early stage of HCC curative treatments, such as tumor resection, ablation, and liver transplantation may offer a cure or prolonged life span, patients with advanced HCC have no effective curative therapy [5]. What is more, the prognosis of HCC patients is extremely poor due to high recurrence, metastasis and chemoresistance rates [6,7]. Sorafenib, the only FDA approved first line treatment for advanced HCC, is a multi-target kinase inhibitor for Raf kinases, vascular endothelial growth factor and platelet-derived growth factor receptors. Sorafenib improves survival with a median OS rate of 6.5 months compared to the placebo group [8,9,10]. However, patients with advanced HCC predominantly develop resistance to sorafenib treatment and their survival benefit is limited to 3–5 months with severe side effects [11]. Thus, developing new effective therapies and identifying novel diagnostic and prognostic biomarkers for HCC are necessary.

Autophagy plays an important role in maintaining cellular and metabolic homeostasis in the liver under physiological conditions [12,13]. Autophagy regulates many metabolic processes, such as glycogenolysis, lipolysis and protein catabolism, which are essential for liver function. Besides, at the cellular level, autophagy maintains organelle homeostasis by modulating mitophagy [14], pexophagy, lipophagy and ER-phagy [15,16]. Moreover, autophagy has been identified as a resistance mechanism to HCC therapy. The protein ULK1 (unc-51 like autophagy activating kinase 1) acts as a key regulator of autophagy [17,18]. ULK1 has been reported to improve the prognostic assessment of HCC [19,20]. In our previous study, we verified that sorafenib treatment induced autophagy and loss of ULK1 could increase sorafenib sensitivity in HCC cells by inhibiting autophagy. We further screened a highly selective ULK1 kinase inhibitor named XST-14, which could block autophagy and induce apoptosis in HCC cells [21]. XST-14 exerted anti-HCC effects and was a good candidate for future investigation.

However in the process of systematic evaluation of drug candidates, including structural evaluation, safety PK and PD and manufacturability et al., we found that XST-14 is challenged by its poor manufacturability and dissolution characteristics. In order to screen small molecule compounds which target ULK1 with better manufacturability, we performed structure-based virtual screening. Based on the pharmacophore 6QAS, 5CI7 and 4WNO of ULK1, we screened 21 k compounds and identified ZZY-19 with better HCC cell killing effects. We next evaluated its role in autophagy and HCC progression and determined that targeting ULK1 in combination with sorafenib treatment could serve as a promising interventional strategy for treating HCC.

## 2. Results and Discussion

### 2.1. Pharmacophore Model Construction for Virtual Screening

We used Receptor-Ligand Pharmacophore Generation module of Discovery Studio (v16.1.0, Dassault Systèmes Biovia Corp, San Diego, CA, USA) to generate pharmacophore models, based on the X-ray structure of UKL1 in complex with ligands (PDB code: 5CI7, 6QAS, 4WNO). The features of the pharmacophore models are shown in Table 1 and Figure 1. For 5CI7, the selectivity score of pharm 1 is higher than pharm 2, so we chose pharm 1, which consists of a hydrogen bond acceptor, a hydrogen bond donor, two hydrophobic features and a cationic feature. According to the binding mode of the ligand to the acceptor, the NH2 of the ligand small molecule acting as a hydrogen bond donor forms a hydrogen bond interaction with Asp99, so we modified the cationic feature to be a hydrogen bond donor. For 6QAS, the selectivity score of pharm 3 is higher than pharm 4, pharm 5, and pharm 6, so we chose pharm 3, which consists of two hydrogen bond acceptors, one hydrogen bond donor, and three hydrophobic features. In order to increase the diversity of compound selection, we also used pharm 4 which has one less critical hydrophobic feature than pharm 3. For 4WNO, we chose pharm 7 with a higher selectivity score than pharm 8. Pharm 7 consists of two hydrogen-bond donors, and two hydrophobic features. According to the ligand-acceptor mode of action, we noted that the pyrazole sp2 N atom in the ligand molecule acts as a hydrogen-bond acceptor with Cys95 producing hydrogen bond interactions. We add a hydrogen bond acceptor feature to the pyrazole sp2N atom based on the pharmacophore.

### 2.2. Molecular Docking Screening of Small Molecules and ULK1 Protein

All the conformers in the 3D database were mapped to the pharmacophore model by a rigid fit algorithm (“FAST” search) implemented in the “Search 3D Database” module in DS. The metric of FitValue was used to measure the similarity of pharmacophore features. All the conformers with FitValues greater than zero were put out. For the conformers/stereoisomers that belong to the same Specs ID, only the conformer/stereoisomer that best matched the pharmacophore model was saved. The computational workflow consisted of pharmacophore filtering, docking and a Lipinsiki Filter. The screening process was shown in Figure 2. We screened the Specs chemical library (~210,000 compounds) for potential UKL1 inhibitors with the computational workflow. As the first step of molecular docking using OpenEye (OpenEye Scientific Software, Inc., Santa Fe, NM, USA), a maximum of 200 conformers was generated by the module named OMEGA (version 2.5.1.4). Secondly, the prepared protein structure was converted to a receptor by OEDocking (version 3.0.1), with the cognate ligand to define the binding site. Thirdly, all the conformers of the compound were positioned in the binding site of the receptor and scored by the Chemgauss4 scoring function in OEDocking. Lastly, the top-scoring pose was retained and used as the initial binding mode between the compound and UKL1. The structural clustering into 15 clusters based on FCFP_6 fingerprints facilitated the selection of structurally diverse compounds. By visual inspection, we selected one or two chemical structures from each cluster by giving priority to those with greater FitValues and Chemgauss4 scoring. The selected compounds were shown in Table 2, Figure 3 and Figure 4.

### 2.3. Validation of ULK1 Inhibitors

We then selected the top 19 compounds and firstly examined the anti-proliferative ability of HepG2 and HCCLM3 cells. Since the compound ZZY-12 is insoluble, we did not choose it in subsequent experiments. We found that the ZZY-19 displayed well cell killing effects (Figure 5A,B), which was better than XST-14. We then examined the binding abilities of these 18 candidate ULK1 inhibitors using surface plasmon resonance (SPR) analysis (Table 3). Among these small molecules, ZZY-19 (KD = 5.466 × 10^−10^ M) displayed a high binding affinity for ULK1 (Figure 5C). Taken together, ZZY-19 were potential candidate compounds that targeted ULK1.

### 2.4. ZZY-19 Inhibits the Proliferation, Invasion, Migration, and Induces Apoptosis of HCC Cells

To further determine the effect of ZZY-19 on HCC cells, we detected the proliferation, invasion, migration, and apoptosis of HepG2 and HCCLM3 cells treated with ZZY-19. We found that ZZY-19 remarkably inhibits the proliferation of HepG2 and HCCLM3 cells. The IC_50_ of ZZY-19 was 54.6 μM in HepG2 and 80.6 μM in HCCLM3 cells (Figure 6A,B). We then performed invasion assays and wound healing assays to evaluate the effects of ZZY-19 on the invasion and migration of HCC cells. Flow cytometry analysis with Annexin V-PI staining was performed to evaluate the percentage of apoptotic cells in HepG2 and HCCLM3 cells treated with ZZY-19. We found that ZZY-19 could decrease the invasion (Figure 6C,D) and migration (Figure 6E,F) of HepG2 and HCCLM3 cells and significantly increased the percentage of apoptotic cells (Figure 6G,H). These results showed that ZZY-19 possessed promising anti-HCC activities.

### 2.5. ZZY-19 Induces Autophagy Inhibition by Reducing the Expression of the ULK1 and Acts Synergistically with Sorafenib

We then investigated the effects of the synergistic effects of ZZY-19 and sorafenib on the proliferation activities of HCC cells. We found that the ZZY-19 remarkably sensitized HepG2 and HCCLM3 cells to sorafenib’s cell killing effects (Figure 7A,B). Western blotting was used to detect the effects of ZZY-19 with or without sorafenib on autophagy related proteins. We found that ZZY-19 treatment decreased LC3-II, Beclin 1, VPS34 and ULK1 levels in HCC cells, and increased soluble and insoluble p62 levels. Furthermore, the combination of ZZY-19 and sorafenib counteracted the autophagy activation of sorafenib, indicated with the lowered LC3-II, Beclin 1, VPS34 and ULK1 and elevated p62-sol and p62-insol levels in ZZY-19 plus sorafenib treated HepG2 and HCCLM3 cells (Figure 7C,D). To further test the ability of ZZY-19 to block autophagy, HepG2 cells were infected with GFP-LC3 adenovirus to monitor autophagy activity. HepG2 cells were treated with ZZY-19 (10 μM), sorafenib (10 μM), or ZZY-19 plus sorafenib for 24 h. We found that ZZY-19 strongly inhibited the conversion of LC3-I to LC3-II and the combination of ZZY-19 and sorafenib showed a marked inhibitory effect. The number of GFP-LC3 puncta in HepG2 cells was dramatically decreased by ZZY-19 or ZZY-19 plus sorafenib treatment (Figure 7E). These data suggested that ZZY-19 inhibits autophagy by reducing ULK1 kinase activity.

### 2.6. The Combination of ZZY-19 with Sorafenib Synergistically Suppresses the Progression of HCC In Vivo

We further examined the antitumor effect of ZZY-19 in cell line-derived xenograft (CDX) models of HepG2 cells. ZZY-19 suppressed the growth and weight of HepG2 tumors, and the combination of ZZY-19 and sorafenib showed a marked inhibitory effect on HepG2 tumor growth and tumor weight (Figure 8A–C). Collectively, these data suggest that ZZY-19 sensitizes HepG2 cells to sorafenib.

## 3. Materials and Methods

### 3.1. Construction of Small Molecule Databases and 3D Conformational Libraries

The Specs chemical library that included more than 210,000 compounds (version June 2019, accessed on 12 January 2021 at http://www.specs.net) was used for the virtual screening. The compounds in the library were prepared by the “Prepare Ligands” module in DS. The ligand preparation included the generation of all protonated states at the pH of 7.4 and the enumeration of all potential stereoisomers. A multi-conformer database of the prepared structures was built by the “Build 3D Database” module in Discovery Studio. In the database, each prepared structure is represented by a maximum of 100 conformers. As the first step, all the conformers in the 3D database were mapped to the pharmacophore model by a rigid fit algorithm (“FAST” search) implemented in the “Search 3D Database” module in DS.

### 3.2. Preparation of the Protein Molecule

The X-ray structure of UKL1 in complex with ligands (PDB code: 6QAS,5CI7, 4WNO) was downloaded from the Protein Data Bank (accessed on 15 December 2020 at https://www.rcsb.org). Then, the identical protein chains and the co-crystallized water molecules were deleted, and the cognate ligand was stripped from the crystal structure and saved for future use. The “Clean Protein” and “Prepare Protein” tools of Discovery Studio were used to solve potential problems in the protein structure, such as nonstandard names, incomplete residues, nonstandard atom orders, alternate conformations, as well as incorrect connectivity and bond orders, modification of all hydrogen atoms and terminal residues, and generate the protonation state at pH 7.0.

### 3.3. Mouse Models for Tumor Growth

A xenograft mouse model was used to evaluate the effect of combination treatment with sorafenib and ZZY-19 on tumor growth. The right flanks of 6-week-old male BALB/c nude mice were inoculated subcutaneously with HepG2 cells (2 × 10^6^) diluted with 100 μL of Matrigel (Corning, 354230) at a 1:1 ratio per mouse. The mice were earmarked randomly separated into four groups (*n* = 8 per group) in a blinded manner. Tumors were allowed to grow for 7 days, and mice were then administered vehicle (Kolliphor^®^ HS 15, i.g., once a day), sorafenib (30 mg/kg, i.g., once a day), ZZY-19 (30 mg/kg, i.p., once a day), or ZZY-19 plus sorafenib for 14 days. Tumor burden was evaluated by measuring tumor volume using the formula: Volume = Width^2^ × Length × 0.5 to determine the therapeutic efficacy in the mouse models.

### 3.4. Cell Culture

All HCC cells were purchased from the China Center for Type Culture Collection (CCTCC). HepG2 cells were grown in MEM (Invitrogen, Carlsbad, CA, USA, 11090081) supplemented with nonessential amino acids (Invitrogen, 11140050). HCCLM3 cells were cultured in DMEM. All media were supplemented with 10% FBS (Invitrogen, CA, USA) and penicillin-streptomycin. All cell lines had recently been authenticated by short tandem repeat (STR), profiling and characterized by mycoplasma detection and cell viability detection.

### 3.5. CCK-8 Assay

For the Cell Counting Kit-8 (CCK-8) assay, HepG2 and HCCLM3 cells were seeded in 96-well plates at a density of 2000 cells/well. For the verification of the ULK1 inhibitors on the proliferation of HepG2 and HCCLM3 cells, the sorafenib (20 μM) and the ULK1 inhibitors (100 μM) treated cells were cultured for 24 h. For the effects of ZZY-19 on the proliferation of HepG2 and HCCLM3 cells, ZZY-19 (3.125, 6.25, 12.5, 25, 50, 100, 200, 300, 400 μM) treated cells were cultured for 24 h. For the effects of ZZY-19 acting synergistically with sorafenib on the proliferation of HepG2 and HCCLM3 cells, DMSO or ZZY-19 (10 μM) with sorafenib (1.5625, 3.125, 6.25, 12.5, 25, 50, 100, 150, 200 μM) treated cells were cultured for 24 h. Subsequently, 100 μL of complete medium supplemented with 10 μL of CCK-8 solution (Dojindo, CK04) was added to each well, and the plates were incubated for 2 h. Finally, the absorbance was measured at 450 nm.

### 3.6. Flow Cytometry

For the apoptosis assays, HepG2 and HCCLM3 cells were treated with ZZY-19 (10 μM), Sorafenib (10 μM). After treatment for 24 h, the cells were harvested and stained with ANXA5/propidium iodide kit. The data was acquired using a FACSCanto II flow cytometer (BD). FCS EXPRESS software (De Novo software, Los Angeles, CA, USA) was used for data analysis.

### 3.7. Surface Plasmon Resonance Analysis

Surface plasmon resonance (SPR) analyses were conducted using the ULK1 protein (OriGene Technologies) immobilized onto channel 2 in a CM5 sensor chip (GE Healthcare, Fairfield, CT, USA) through a standard coupling protocol with the BIAcore S200 system (GE Healthcare). To measure the binding kinetics, ZZY-19 (10~0.3125 nM in 2-fold serial dilutions) and a buffer blank for baseline subtraction were sequentially injected, with a regeneration step (glycine, pH 2.5) performed between each cycle. The equilibrium dissociation constant was calculated with BIA evaluation software.

### 3.8. Autophagosomes Assay

HepG2 cells were infected with the GFP-LC3 adenovirus. After 24 h, the cells were starved in Earle’s balanced salt solution (EBSS, Thermo Fisher Scientific, Waltham, MA, USA, 14155063) for 4 h and treated with ZZY-19 (10 μM), sorafenib (10 μM), or ZZY-19 plus sorafenib for 24 h. Then, the GFP-LC3 was observed by live cell imaging microscopy.

### 3.9. Invasion Assays

Invasion assays were performed using transwell chambers with filter membranes with an 8 μm pore size (Millipore, SCWP04700). The chambers were precoated with 10 μg/mL FN1 (fibronectin 1; MERCK, F0635) on the lower surface, and the polycarbonate filters were coated with matrigel (40 μL per well, BD Matrigel Matrix, 354230). Thereafter, the chambers were inserted into 24-well culture plates. Single-cell suspensions were seeded into the upper chamber (5 × 10^4^ cells per well in 0.4% FBS in DMEM); 12 h later, the non-invaded cells on the upper side of the filter were removed with a cotton swab. The invaded cells were fixed with 4% paraformaldehyde in PBS, stained with 0.1% crystal violet and counted in six random fields using brightfield microscopy.

### 3.10. Wound Healing Assays

HepG2 and HCCLM3 cells were harvested and seeded into 12-well plates at 2 × 10^5^ cells per well. When the cells were of 90% confluence, the cells were scratched with a 200-μL pipette tip to create a wound and washed with PBS to remove the suspended cells. The cells were treated with ZZY-19 (10 μM) or Sorafenib (10 μM), respectively, for 24 h. The cells were imaged with an inverted microscope (Olympus, Tokyo, Japan). The scratch healing efficiency was analyzed by calculating the distance of the wound.

### 3.11. Western Blotting Analysis

The cells were lysed with RIPA lysis buffer (Cell Signaling Technology, MA, USA) supplemented with PMSF on ice. Then the cell lysates were centrifuged at 12,000 rpm. The BCA protein assay kit (Beyotime Biotechnology, Nantong, Jiangsu, China) was used to quantify the protein content; 30 μg of protein were subjected to SDS-PAGE and transferred onto PVDF membranes (Millipore, Boston, MA, USA) for immunoblotting analysis. Images of the Western blots were acquired with a Tanon 5200 chemiluminescent imaging system (Tanon, Shanghai, Beijing).

### 3.12. Statistical Analysis

The statistical analyses were performed using GraphPad Prism 8.3 software. The data are representative and/or the means ± SEM of three assays. An unpaired two-sided Student’s *t*-test was used to compare the mean values of the two groups. One-way ANOVA was used to compare multiple groups. Generally, all experiments were performed with n ≥ 3 biological replicates. *p* < 0.05 was considered statistically significant.

## 4. Conclusions

In this study, we first generated pharmacophore models of ULK1 based on the X-ray structure of UKL1 in complex with ligands and screened the potential UKL1 inhibitors. Through CCK8 activity screening and SPR assays, we found that ZZY-19 displayed a higher binding affinity for ULK1. Moreover, ZZY-19 also performed well in anti-HCC activities. Especially, ZZY-19 induces autophagy inhibition by reducing the expression of the ULK1. The combination of ZZY-19 with sorafenib synergistically suppresses the progression of HCC in vivo. Taken together, ZZY-19 was a potential candidate compound that targeted ULK1 and possessed promising anti-HCC activities by inhibiting autophagy. Our study provides a potential strategy for the treatment of liver cancer.

## Figures and Tables

**Figure 1 molecules-27-02627-f001:**
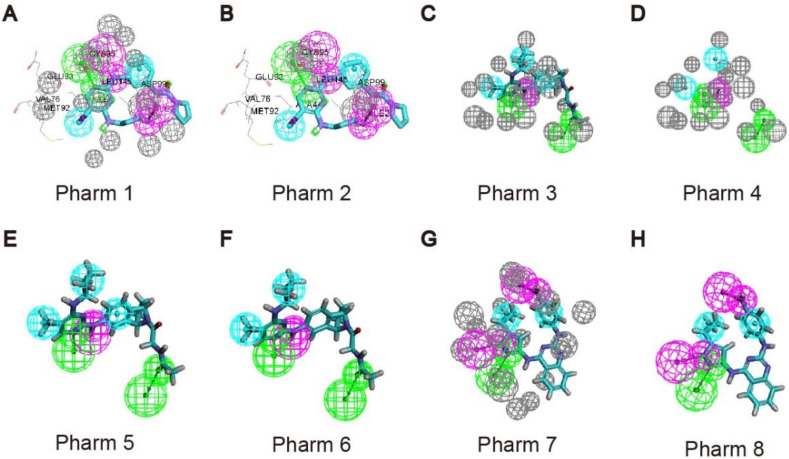
Generate pharmacophore models of UKL1 in complex with ligands. (**A**,**B**) Generate pharmacophore models of 5CI7; (**C**–**F**) Generate pharmacophore models of 6QAS; (**G**,**H**) Generate pharmacophore models of 4WNO.

**Figure 2 molecules-27-02627-f002:**
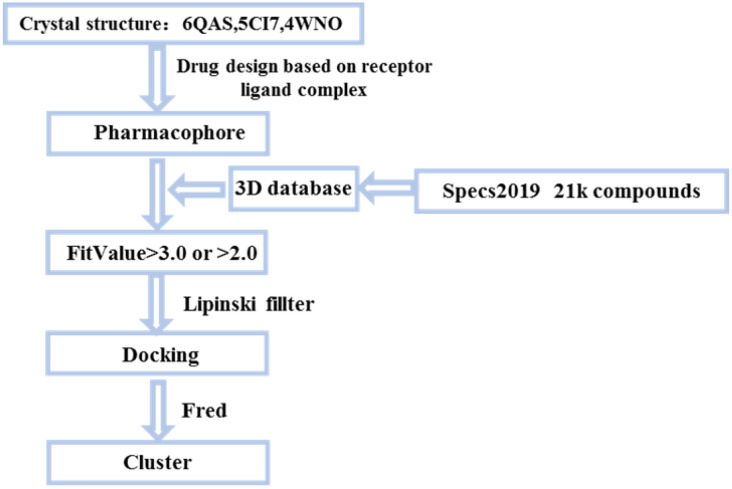
Small molecules screening process.

**Figure 3 molecules-27-02627-f003:**
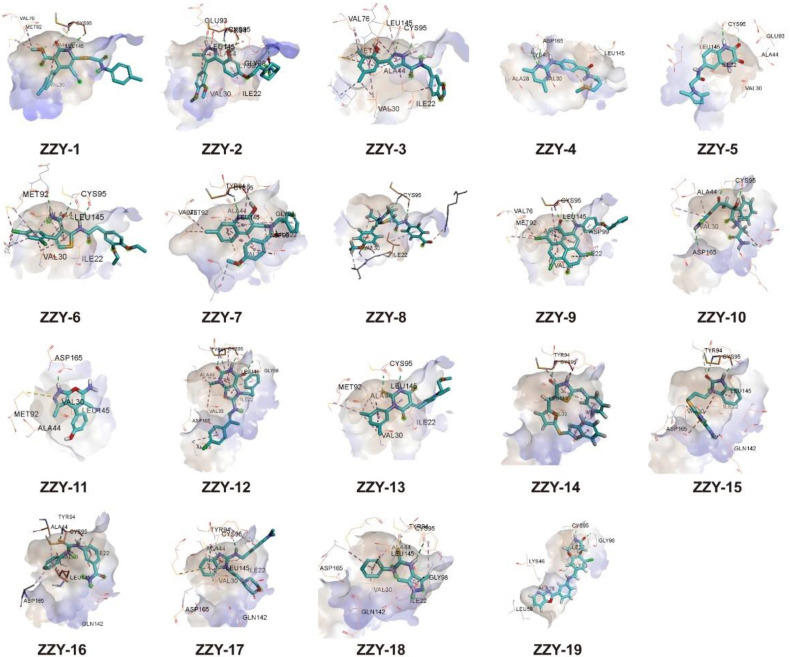
Binding pattern of selected compounds to ULK1 protein.

**Figure 4 molecules-27-02627-f004:**
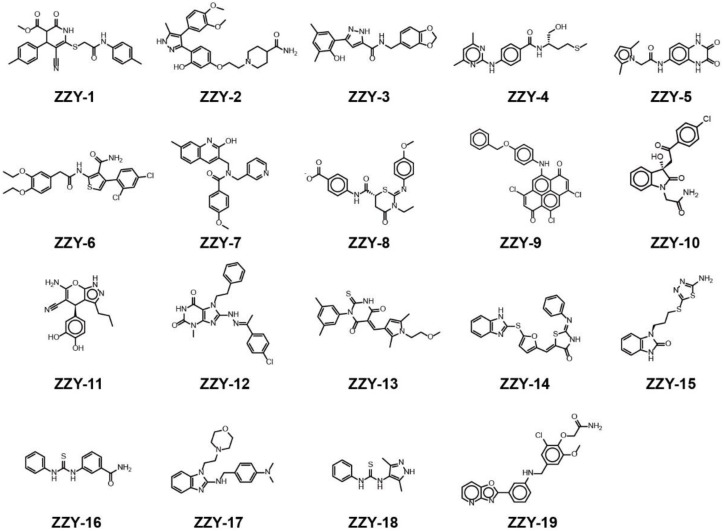
The chemical structures of ZZY-1 to ZZY-19.

**Figure 5 molecules-27-02627-f005:**
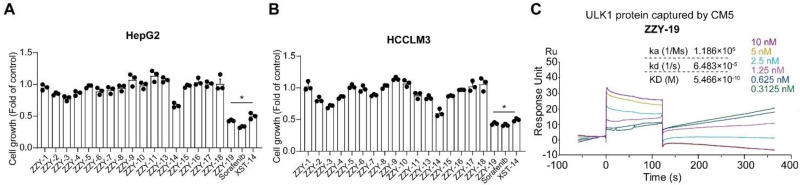
Structure-based virtual screening for the discovery and validation of ULK1 inhibitors (**A**,**B**) Effects of the ULK1 inhibitors on the proliferation of HepG2 and HCCLM3 cells. The inhibition rates were calculated by comparing with the control group; (**C**) The kinetic interactions of the ULK1 protein and ZZY-19 were determined by SPR analyses. The indicated concentrations of ZZY-19 were passed over immobilized ULK1 on CM5 sensor chips. The affinity constants were evaluated using BIA evaluation software. * *p* < 0.05.

**Figure 6 molecules-27-02627-f006:**
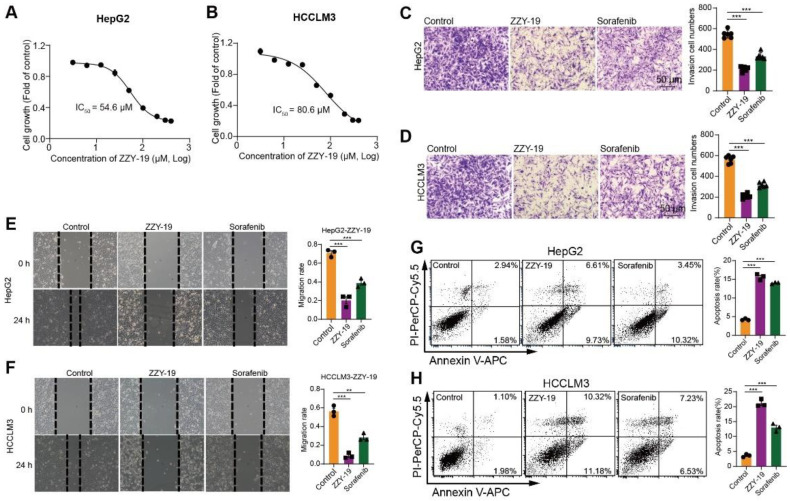
ZZY-19 inhibits the proliferation, invasion, and migration and induces apoptosis of HCC cells. (**A**,**B**) Effects of ZZY-19 on the proliferation of HepG2 and HCCLM3 cells. The data showed a summary of the IC_50_ values for ZZY-19 on HepG2 or HCCLM3 cells. (**C**,**D**) Representative invasion images and quantification of HepG2 and HCCLM3 cells. HepG2 and HCCLM3 cells were treated with ZZY-19 (10 μM), Sorafenib (10 μM). Scale bar, 50 μm. (**E**,**F**) Representative migration images and quantification of HepG2 and HCCLM3 cells. HepG2 and HCCLM3 cells were treated with ZZY-19 (10 μM), Sorafenib (10 μM). Scale bar, 50 μm. (**G**,**H**) Representative flow cytometry images of Annexin V-PI staining and quantification of apoptosis. HepG2 and HCCLM3 cells were treated with ZZY-19 (10 μM), Sorafenib (10 μM). After treatment for 24 h, apoptotic cells were evaluated via Annexin V-PI staining (left), and the histograms showed statistical results of apoptosis rates (right). ** *p* < 0.01; *** *p* < 0.001.

**Figure 7 molecules-27-02627-f007:**
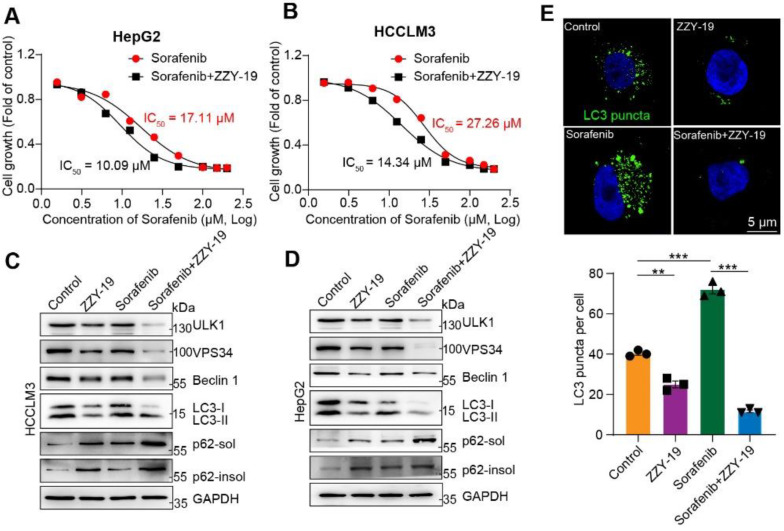
ZZY-19 induces autophagy inhibition by reducing the expression of the ULK1 and acts synergistically with sorafenib (**A**,**B**) Effects of ZZY-19 on the proliferation of HepG2 and HCCLM3 cells. The data showed a summary of the IC_50_ values for sorafenib in HepG2 or HCCLM3 cells. (**C**,**D**) The effects of ZZY-19 and/or sorafenib on the proteins related to autophagy. HepG2 and HCCLM3 cells were treated with ZZY-19 (10 μM), sorafenib (10 μM), or ZZY-19 plus sorafenib. The proteins were detected by immunoblotting. Data are representatives of three independent assays. (**E**) Representative images of HepG2 cells infected with GFP-LC3 adenovirus treatment with ZZY-19 (10 μM), sorafenib (10 μM), or ZZY-19 plus sorafenib for 24 h. Scale bar, 5 μm. ** *p* < 0.01; *** *p* < 0.001.

**Figure 8 molecules-27-02627-f008:**
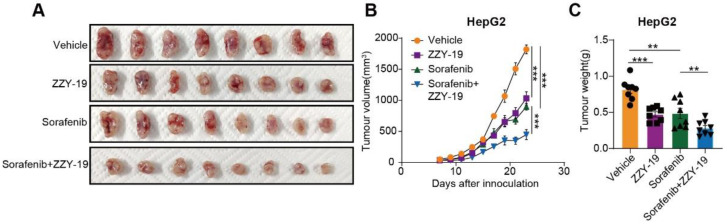
The combination of ZZY-19 with sorafenib synergistically suppresses the progression of HCC in vivo. (**A**) Representative images of tumors in nude mice. The mice were injected subcutaneously with HepG2 cells (*n* = 8). Seven days after injection, the mice were oral gavage with vehicle (Kolliphor^®^ HS 15, i.g., once a day), sorafenib (30 mg/kg, i.g., once a day), ZZY-19 (30 mg/kg, i.p., once a day), or ZZY-19 plus sorafenib for 14 days. (**B**) Effects of the indicated treatments on tumor growth in HepG2 CDX mouse models. (**C**) Effects of the indicated treatments on tumor weight in HepG2 CDX mouse models. ** *p* < 0.01; *** *p* < 0.001.

**Table 1 molecules-27-02627-t001:** Scoring of selected pharmacophore.

Pharmacophore	PDB Code	Number of Fratures	Feature Set	SELECTIVITY SCORE
Pharm 1	5CI7	5	ADHHP	9.6401
Pharm 2	5CI7	4	DHHP	8.1253
Pharm 3	6QAS	6	AADHHH	10.104
Pharm 4	6QAS	5	AADHH	8.5865
Pharm 5	6QAS	5	AADHH	8.5865
Pharm 6	6QAS	5	AADHH	8.5865
Pharm 7	4WNO	4	DDHH	7.8511
Pharm 8	4WNO	4	ADHH	6.9375

PDB: protein data bank.

**Table 2 molecules-27-02627-t002:** Scoring of selected compounds.

Name	NO.	Chemguass4	FitValue	Cluster
ZZY-15	AI-204/31685060	−7.44662	3.99539	9
ZZY-11	AM-807/37225039	−9.59389	3.92534	16
ZZY-10	AK-778/41182442	−9.03607	3.90612	10
ZZY-12	AG-690/11571846	−11.9452	3.72307	3
ZZY-17	AE-848/37018081	−7.39923	3.60915	6
ZZY-13	AO-081/15386942	−10.6991	3.57622	3
ZZY-4	AO-022/43512437	−9.56718	3.46083	14
ZZY-19	AN-465/43369862	−10.7232	3.37745	2
ZZY-2	AO-022/43452377	−11.2999	3.36159	16
ZZY-7	AO-022/43246495	−8.86233	3.30666	22
ZZY-14	AO-081/14338084	−8.26256	3.2225	23
ZZY-1	AO-022/40938770	−8.62358	3.21285	18
ZZY-5	AO-022/43513820	−8.44326	3.21113	11
ZZY-3	AF-399/43343599	−9.70447	3.20811	22
ZZY-16	AN-329/40278680	−7.5261	3.20423	11
ZZY-18	AN-465/40769708	−7.51775	3.16531	20
ZZY-9	AF-399/37112065	−10.985	3.11801	3
ZZY-6	AK-968/41923356	−8.51722	3.08722	6
ZZY-8	AG-690/15441762	−7.71729	3.00193	3

**Table 3 molecules-27-02627-t003:** Surface test of the small molecules on ULK1 captured by CM5.

Name	Mol. wt.	Surface Test-RelResp		RU	Concentration (nM)	NO.
		Baseline (Ru)	Binding (Ru)			
ZZY-1	449.53	0.0	22.9	22.9	100	9
ZZY-2	480.56	0.2	10.0	9.8	100	14
ZZY-3	365.39	0.8	3.8	3.0	100	16
ZZY-4	360.47	−10.0	6.9	16.9	100	10
ZZY-5	312.33	−4.1	33.6	37.7	100	5
ZZY-6	493.40	−6.7	−7.0	−0.3	100	18
ZZY-7	413.48	−3.6	39.7	43.3	100	3
ZZY-8	426.47	−2.1	23.4	25.5	100	7
ZZY-9	532.80	0.9	1.7	0.8	100	17
ZZY-10	411.52	−3.3	51.5	54.8	100	2
ZZY-11	312.33	4.3	19.5	15.2	100	12
ZZY-13	358.78	−5.5	19.5	25.0	100	8
ZZY-14	418.49	2.7	193.6	190.9	100	1
ZZY-15	307.39	−2.9	38.2	41.1	100	4
ZZY-16	271.34	3.5	10.3	6.8	100	15
ZZY-17	379.51	−7.1	6.4	13.5	100	13
ZZY-18	246.33	−9.7	6.4	16.1	100	11
ZZY-19	438.87	−5.9	25.0	30.9	100	6

## Data Availability

Data will be made available upon request.

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
