# Peer review of "Structure-Based Virtual Screening towards the Discovery of Novel ULK1 Inhibitors with Anti-HCC Activities"

_molecules, 2022, doi:10.3390/molecules27092627_

Round 1
Reviewer 1 Report
The article entitled “Structure-based virtual screening towards the discovery of novel ULK1 inhibitors with anti-HCC activities” described a virtual screening study of ULK1 inhibitors with anti-HCC activity. The best inhibitor was tested in cell lines and several experimental techniques were performed to evaluate its inhibition efficacy.
The manuscript is overall well-written and clear. Therefore, I recommend its publication after some minor revisions:
- Figure 1: Poor figure quality makes it almost impossible to understand the compounds’ structures, particularly for Pharm4. A 2D figure with all the pharmacophores would also be very useful.
- Section 2.1.: The authors did not provide a proper discussion of pharmacophores 2, 5, 6, and 8.
- Table 1: The feature set should be provided with a proper caption assessing the meaning of each letter since it could be unknown to some researchers.
- Table 2: The results could be presented sorted by their score to ease the analysis.
- Figure 3: This figure has really poor quality. It needs to be improved. Moreover, the authors should provide, in SI, the structural files of all poses for inspection. Computational results need to be provided in SI or a repository for peer-review and future reference.
- The conclusions could be improved to highlight the main results obtained for the ZZY-19 molecule.
Author Response
Response to Reviewer 1 Comments
The article entitled “Structure-based virtual screening towards the discovery of novel ULK1 inhibitors with anti-HCC activities” described a virtual screening study of ULK1 inhibitors with anti-HCC activity. The best inhibitor was tested in cell lines and several experimental techniques were performed to evaluate its inhibition efficacy.
The manuscript is overall well-written and clear. Therefore, I recommend its publication after some minor revisions:
- Figure 1: Poor figure quality makes it almost impossible to understand the compounds’ structures, particularly for Pharm4. A 2D figure with all the pharmacophores would also be very useful.
Re: Following your suggestion, we have replaced Figure 1 with better quality.
2. Section 2.1.: The authors did not provide a proper discussion of pharmacophores 2, 5, 6, and 8.
Re: Following your suggestion, we have modified the description of the principle of our choice. According to the selectivity score, pharm 1 is higher than pharm 2 for 5CI7, pharm 3 is higher than pharm 4, pharm 5, and pharm 6 for 6QAS, and pharm 7 is higher than pharm 8 for 4WNO. We have supplemented these descriptions in Section 2.1 of revised manuscript (page 2).
3. Table 1: The feature set should be provided with a proper caption assessing the meaning of each letter since it could be unknown to some researchers.
Re: Thank you for your careful observation. Following your suggestion, we have provided the explanation of PDB below Table 1 (page 2). PDB: protein data bank.
4. Table 2: The results could be presented sorted by their score to ease the analysis.
Re: Thank you for your professional criticism. We have sorted the results by their score in the revised manuscript (page 4).
5. Figure 3: This figure has really poor quality. It needs to be improved. Moreover, the authors should provide, in SI, the structural files of all poses for inspection. Computational results need to be provided in SI or a repository for peer-review and future reference.
Re: Following your suggestion, we have replaced Figure 3 with better figure quality. We have exhibited the chemical structures of ZZY-1 to ZZY-19 in revised Figure 4 (page 6). The computational results have been provided in attachment 1.
6. The conclusions could be improved to highlight the main results obtained for the ZZY-19 molecule.
Re: Following your suggestion, we have modified the description of the conclusion in the revised manuscript as follows: Not only does our study point out that ZZY-19 possessed promising anti-HCC activities by inhibiting autophagy, but also stressed that ZZY-19 plus sorafenib provides a potential strategy for the treatment of liver cancer.
Reviewer 2 Report
In this manuscript, the authors have identified a series of novel classes of ULK1 inhibitors and found that the compound named ZZY-19 is a more promising inhibitor than the previously described inhibitor XST-14 to suppress HCC progression. Initially, the authors have identified a series of compounds (ZZY-1 to ZZY-19) using the Receptor-Ligand Pharmacophore Generation module to generate pharmacophore models based on the X-ray structure of ULK1 in complex with ligands (PDB code: 5CI7, 6QAS, 4WNO), followed by molecular docking screening of small molecules and ULK1 protein. From these 19 compounds, ZZY-19 was identified as the lead compound by determining their anti-proliferative ability against HepG2 and HCCLM3 cells. Further, the authors have treated the ZZY-19 with HepG2 and HCCLM3 cells to determine the proliferation, invasion, migration, and apoptosis of treated cells with ZZY-19. Finally, the authors have utilized the western blot analysis to determine the synergistic effects of ZZY-19 and sorafenib on the proliferation activities of HCC cells and found that ZZY-19 treatment decreased LC3-II, Beclin 1, VPS34, and ULK1 levels in HCC cells, and increased soluble and insoluble p62 levels. Overall this is a systematic study for the discovery of a novel class of ULK1 inhibitors to treat Hepatocellular Carcinoma. Here I am requesting one minor revision which is please include one figure in the manuscript showing the chemical structures of ZZY-1 to ZZY-19.
Author Response
Re. Following your suggestion, we have exhibited the chemical structures of ZZY-1 to ZZY-19 in Figure 4.